# Continuous Hydrogen Production via Hydrothermal Gasification of Biodiesel Industry Wastewater: Experimental Optimization and Energy Integration Simulation

Isabela R. Teixeira [1], Isabela M. Dias [1], Lucas C. Mourão [1], Laiane A. Andrade [2], Leandro V. Pavão [3], Jose M. Abelleira-Pereira [4], Guilherme B. M. Souza [1], Lucio Cardozo-Filho [3], Christian G. Alonso [1] and Reginaldo Guirardello [5,*]

[1] Instituto de Química, Universidade Federal de Goiás (UFG), Av. Esperança s/n, Campus Samambaia, CEP, Goiânia 74690-900, GO, Brazil; isabelarodrigues652@gmail.com (I.R.T.); isabelamilhomem@ufg.br (I.M.D.); lucasclementinom@gmail.com (L.C.M.); guilherme_botelho@ufg.br (G.B.M.S.); christian@ufg.br (C.G.A.)
[2] Production Engineering Faculty, Universidade Federal do Mato Grosso do Sul (UFMS), Av. Rosilene Lima Oliveira, 64, Jardim Universitário, CEP, Nova Andradina 79750-000, MS, Brazil; laiane.andrade@ufms.br
[3] Programa de Pós-Graduação em Engenharia Química, Universidade Estadual de Maringá (UEM), Avenida Colombo, 5790-Zona 7, Maringá 87020-900, PR, Brazil; lvpavao2@uem.br (L.V.P.); lcfilho@uem.br (L.C.-F.)
[4] Department of Chemical Engineering and Food Technology, Faculty of Sciences, University of Cádiz, International Excellence Agrifood Campus (CeiA3), Puerto Real, 11510 Cádiz, Spain; jose.abelleira@uca.es
[5] School of Chemical Engineering, University of Campinas (UNICAMP), Av. Albert Einstein 500, Campinas 13083-852, SP, Brazil
* Correspondence: guira@feq.unicamp.br

**Abstract:** This study reports the continuous production of $H_2$ from the wastewater effluent of the biodiesel industry in a medium containing water under supercritical conditions. The effects of temperature and feed flow rate on the generation of hydrogen were evaluated and optimized. At a temperature of 700 °C and a 17.5 mL/min feed flow, a total gas flow of 5541 NmL/min was achieved. Among all identified gases, hydrogen represented the highest molar fraction of 73%. Under optimized conditions, a $H_2$ yield of 357 NmL/$g_{effluent\ feed}$ was observed. The experimental results indicate a significant increase in the $H_2$ production at the highest experimented temperatures. On the other hand, the feed flow only slightly influenced the process within the assessed range but showed a tendency to increase the $H_2$ production at the highest values. Finally, information on energy efficiency optimization and scale-up are presented, and at the same time, different designs for industrial implementation of the hydrothermal gasification process are proposed.

**Keywords:** hydrogen; hydrothermal gasification; supercritical water; wastewater treatment; biodiesel

## 1. Introduction

The growing energy demand witnessed nowadays stems from how industrial production processes have developed since the 1970s with the new industrial revolution. This high demand provoked a sudden increase in the production of fuels, which have been supplied since then, mainly by fossil fuels. Thus, the indiscriminate use of petroleum as an energy source has resulted in the emission of large amounts of polluting compounds, such as $CO_2$ and $CH_4$, into the atmosphere [1–3].

$CO_2$ is one of the main factors responsible for raising the temperature of the planet, a phenomenon called global warming [4]. This effect has already caused significant changes on a global scale, such as the increase in average surface temperature, a disturbance in precipitation regimes, the melting of the polar ice caps, and, consequently, a rise in the sea level [5,6]. Due to perceptible climate changes, the importance of transitioning from energy sources based on fossil fuels to clean energies, which would reduce the emission of greenhouse gases, has been thoroughly discussed.

Considering the current sources of energy supply, in 2022, fossil fuels represented about 60.93% of the total generation, while renewable and nuclear sources accounted for 29.92% and 9.15%, respectively [7]. However, it is already understood that society is facing turbulence in the oil industry, marked by constant instability in product prices, showing signs of an emergent energy crisis [8]. This scenario reveals the necessity of changes in the global energetic commercial structure, and a gradual transition from oil-based fuels to renewable sources is expected [9]. In this sense, recent studies indicate that hydrogen ($H_2$), which is considered a clean and sustainable energy source, has the capacity to complement the growing energy demand and aid the global energetic transition [10–12].

According to Rodriguez Correa and Kruse (2018), several methods are widely used to obtain hydrogen [13]. The more traditional methods are based on techniques such as steam reforming, pyrolysis, and partial oxidation [14–16]. Additionally, biological and electrochemical routes have also been studied as alternative methods for hydrogen production [17]. More recently, the use of supercritical water gasification (SCWG), which consists of the hydrothermal gasification of compounds in a medium containing water at a temperature and pressure above its critical point, is growing rapidly. To reach supercritical conditions, water must surpass the temperature of 373.95 °C and pressure of 22.1 MPa. At that point, water becomes a fluid with specific characteristics that are ideal for conducting reactions, especially highly hydrogen-selective gasification of organic compounds [18,19]. Under supercritical water (SCW) conditions, it is possible to use biomass as a renewable feedstock material to produce hydrogen [4]. Industrial by-products or even waste can be considered attractive organic matter sources for hydrogen production [20–23]. Zoppi and co-workers (2022) reported a carefully reviewed study using an aqueous phase reforming process for the valorization of wastewater streams. Nonetheless, low carbon conversion and total organic carbon degradation were observed. Such issues could be overcome by using both higher pressure and temperature conditions [24]. Therefore, the possibility of using different raw materials, coupled with the high reaction velocity and conversion, highlights the SCWG potential in comparison to other alternative techniques.

As reported by Rocha et al. (2021) [25], several studies assessed the SCWG of glycerol for hydrogen production, which is the main by-product of the transesterification process in a biodiesel production plant. Alongside the energetic potential, the proposition of a valorization route for a compound otherwise considered a residual material is regarded as a key sustainable advantage of the SCW process. Thus, the SCWG of glycerol is viewed as a promising alternative for the sustainable production of $H_2$ [19,26–28].

Although many studies have been conducted aiming at the SCW gasification of glycerol for hydrogen generation, few works have focused on the real effluent generated during the biodiesel production process [29]. Using raw effluent as a feedstock material for hydrogen production exhibits many environmental and economic benefits, such as utilizing an unwanted substrate, reducing treatment costs and carbon emissions, and generating gases with high energetic potential. Thus, the current study investigated the continuous production of hydrogen from effluents collected at a local biodiesel industry in a medium containing water under supercritical conditions. The experiments were performed according to a central composite design, the effects of temperature and feed flow rate on the production of hydrogen and/or synthesis gas were evaluated, and the parameters were optimized. Finally, information on the scalability of the SCW processing of the biodiesel industry wastewater was provided, and two different superstructure designs were simulated and proposed, considering energy efficiency aspects.

## 2. Materials and Methods

### 2.1. Influent Sample

The experimental runs were conducted using wastewater samples collected at a local biodiesel industry located in the Distrito Agroindustrial de Anápolis (DAIA), Anápolis (Brazil). At this particular industrial site, the main raw material used to produce biodiesel is soy oil, which is extracted by a crushing mechanism. Although the biodiesel production

process itself requires a relatively small quantity of water, a significant volume of water is required for washing and cooling the machinery/equipment, and, therefore, a high amount of wastewater is generated from the supporting sub-processes.

### 2.2. Reaction System

The hydrothermal gasification process was carried out in a continuous flow tubular reactor (Inconel 625 alloy) with a length of 30 cm and an internal diameter of 1.1 cm. The reaction system was previously reported in detail by Ribeiro et al. (2021) and Dias et al. (2023) [30,31], as can be seen in Figure 1. In short, the feedstock sample was pumped into the system by a high-pressure isocratic pump (Agilent Technologies 1260). First, the feedstock solution was preheated in a split oven up to 350 °C, which ensures that, upon entering the reactor, it readily reached the supercritical conditions. Next, another split furnace resistance system was used to keep the tubular reactor at the desired temperature since the hydrothermal gasification process is an endothermic process, in which the gasification process with supercritical water and the dissolution/breakdown of the organic matter takes place. The system pressure was maintained constant at 25.0 MPa by a back-pressure regulator and monitored using a pressure gauge. After passing through the reactor, the mixture of liquid and gaseous products was cooled in a helical heat exchanger, refrigerated by a thermostatic bath, and transferred to a phase separator. Finally, the liquid and gaseous phase products were recovered and collected for characterization analysis. The gas flow rates were measured using a soap film bubble meter, and the effluent flow rates correspond to the difference between the initial and final weight of liquid accumulated downstream of the process over a period.

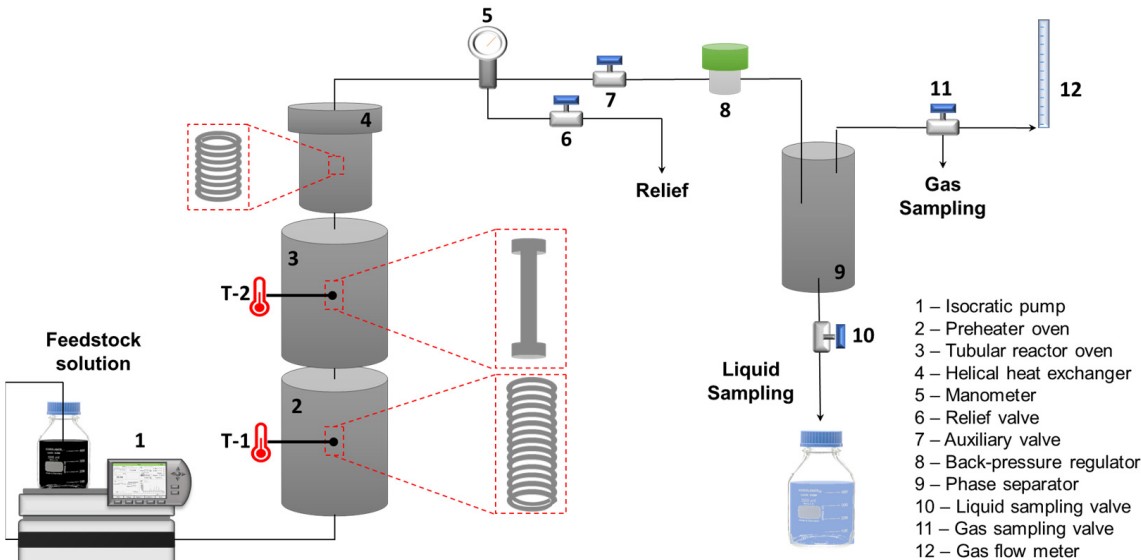

**Figure 1.** Schematic representation of the SCW reaction unit. Adapted from Dias (2023). Reprinted with permission from Ref. [31], 2023, Elsevier.

### 2.3. Chemical Characterization

The feedstock and effluent samples were characterized (as both liquid and gaseous products). The following physicochemical parameters were analyzed in the effluent liquid phase: total organic carbon (TOC), nitrogenous compounds, chemical oxygen demand (COD), biochemical oxygen demand (BOD), oils and greases (OG), sulfates, ammoniacal nitrogen, surfactants, phosphorus, and metals.

The measurement of metals was carried out using optical emission spectrometry with an inductively coupled plasma method (model 7300 DV, Perkin Elmer, MA, USA). Nitrogen compounds (nitrite and nitrate) were measured using molecular absorption spectroscopy in the UV-VIS region (model 365, Perkin Elmer, MA, USA). BOD and COD analyses were

performed using the 5220C method and an optical oximeter (model HQ40D, Hach, IA, USA). The TOC analysis was performed in a carbon analyzer, model TOC-L CSH (Shimadzu, Kyoto, Japan). The oil and grease parameters were determined using a Soxhlet extractor. All analyses were performed in accordance with the methods reported in the 23rd edition of the *Standard Methods for the Examination of Water and Wastewater* [32].

The gaseous product was analyzed by gas chromatography (GC), model Clarus 580 (Perkin Elmer), using a ZB-5MS column equipped with a thermal conductivity detector (TCD) and a flame ionization detector (FID). The analyses were performed at 60 °C using argon as carrier gas at constant flow for 15 min.

### 2.4. Experimental Conditions and Central Composite Design

Aiming to enhance hydrogen production, the main process parameters, feed flow rate (intrinsically related to residence time, internal volume of reactor equals 33 mL), and temperature were optimized. For this purpose, a central composite design (CCD) and a response surface methodology (RSM) were used. These statistical methods enable the determination of the curvature of a linear and the quadratic coefficients of the regression model using two main parameters, one with a group of axial points that provide the curvature estimation and one that gives the variance of equidistant points from the design center, minimizing the variation in the regression coefficients, called orthogonal or rotational coefficient [33]. The experimental conditions were outlined by Statistica™ software version 12. The experimental conditions, determined by CCD, considered two distinct levels, each one associated with two variables resulting in a total of 4 factorial points. Additionally, 4 axial points ($\alpha = \pm 1.41$) were included, and the central point was replicated 4 times. A total of 12 experimental runs were conducted, as shown in Table 1. The selection of the maximum and minimum parameter values was decided according to the operational limitations of the SCW reactor system.

**Table 1.** Experimental conditions outlined by Statistica™ software version 12 according to the central composite design.

| Independent Variables | Levels | | | | |
|---|---|---|---|---|---|
| | −1.41 | −1.00 | 0 | 1.00 | 1.41 |
| Feed flow rate (mL/min), $X_1$ | 10 | 12.2 | 17.5 | 28.8 | 25 |
| Temperature (°C), $X_2$ | 500 | 529 | 600 | 670 | 700 |
| **Run** | **Feed flow rate (mL/min)** | **Temperature (°C)** | | $X_1$ | $X_2$ |
| 1 | 12.2 | 529 | | −1.00 | −1.00 |
| 2 | 12.2 | 670 | | −1.00 | 1.00 |
| 3 | 22.9 | 529 | | 1.00 | −1.00 |
| 4 | 22.9 | 670 | | 1.00 | 1.00 |
| 5 | 10.0 | 600 | | −1.41 | 0.00 |
| 6 | 25.0 | 600 | | 1.41 | 0.00 |
| 7 | 17.5 | 500 | | 0.00 | −1.41 |
| 8 | 17.5 | 700 | | 0.00 | 1.41 |
| 9 | 17.5 | 600 | | 0.00 | 0.00 |
| 10 | 17.5 | 600 | | 0.00 | 0.00 |
| 11 | 17.5 | 600 | | 0.00 | 0.00 |
| 12 | 17.5 | 600 | | 0.00 | 0.00 |

The experimental results of hydrogen production and total organic carbon removal were parameterized by mathematical regression and fitted into a quadratic model. The

parameter's significance was determined by an analysis of variance (ANOVA) using Statistica™ software version 12. The variables were evaluated considering their effects on the selected response with a 95% interval confidence level, $p < 0.05$.

### 2.5. Scale-Up Process Simulation and Energy Efficiency Analysis

The scalability of the biodiesel wastewater processing unit using supercritical water, considering the main energy-related aspects, was performed using Aspen Plus™ software (V9). For this purpose, the species considered during the simulation of the SCW biodiesel wastewater processing were based on the effluent characterized by the work of Yu et al. (2021) [34]. The thermodynamic properties were calculated using the NRTL-HOC package. The components missing from the Aspen Plus database had their properties estimated from their molecular structures using the group contribution method. The RGibbs reactor model was used at 26.0 MPa and 700 °C, the ideal experimental conditions for hydrogen production, and a mass flow rate of 1000 kg/h was considered.

### 3. Results and Discussion

#### 3.1. SCWG of Biodiesel Industry Wastewater

The gaseous product mains, determined by gas chromatography and the total gas flow rate, are shown in Table 2. Hydrogen ($H_2$), carbon dioxide ($CO_2$), carbon monoxide (CO), methane ($CH_4$), ethane ($C_2H_6$), and ethene ($C_2H_4$) were identified. For all experimental conditions assessed, $H_2$ was the major component of the gaseous products.

**Table 2.** Composition and total flow rate of the gas effluent obtained from the supercritical water gasification (SCWG) of effluent samples from the biodiesel production process.

| Run | T (°C) | F (mL/min) | Molar Fraction (% mol/mol) | | | | | | Total Gaseous Flow Rate (mL/min) |
|-----|--------|------------|-------|--------|-------|--------|--------|--------|------|
| | | | $H_2$ | $CH_4$ | CO | $CO_2$ | $C_2H_4$ | $C_2H_6$ | |
| 1 | 529 | 12.3 | 65.89 | 0.88 | 10.76 | 22.24 | 0.13 | 0.11 | 818 |
| 2 | 670 | 12.3 | 71.29 | 1.83 | 4.33 | 22.09 | 0.03 | 0.44 | 3332 |
| 3 | 529 | 22.9 | 62.71 | 0.90 | 13.77 | 22.39 | 0.13 | 0.08 | 282 |
| 4 | 670 | 22.9 | 71.95 | 1.48 | 4.74 | 21.47 | 0.08 | 0.28 | 4927 |
| 5 | 600 | 10.0 | 68.59 | 2.09 | 7.42 | 21.32 | 0.15 | 0.42 | 1271 |
| 6 | 600 | 25.0 | 69.60 | 1.42 | 9.16 | 19.47 | 0.21 | 0.15 | 1576 |
| 7 | 500 | 17.5 | 66.46 | 0.61 | 12.36 | 20.45 | 0.08 | 0.04 | 182 |
| 8 | 700 | 17.5 | 73.86 | 1.82 | 4.54 | 19.33 | 0.06 | 0.39 | 5541 |
| 9 | 600 | 17.5 | 68.80 | 1.81 | 9.94 | 18.96 | 0.22 | 0.26 | 1260 |
| 10 | 600 | 17.5 | 72.84 | 1.42 | 4.92 | 20.47 | 0.17 | 0.19 | 1373 |
| 11 | 600 | 17.5 | 66.28 | 1.62 | 9.60 | 22.06 | 0.21 | 0.23 | 1514 |
| 12 | 600 | 17.5 | 71.44 | 1.33 | 4.87 | 22.04 | 0.15 | 0.16 | 1639 |

Among the experiments performed, run 8, conducted at a feed flow rate of 17.5 mL/min and a temperature of 700 °C, showed the best condition for hydrogen production and achieved a yield of 73.86%. Proportionally, at all evaluated conditions, $CO_2$ showed the second highest gaseous yield, achieving a maximum molar fraction of 22.4% at a feed flow rate of 22.9 mL/min and temperature of 529 °C (run 3). Regarding the studied range, it was observed that a greater $H_2$ selectivity occurred at high temperatures, while higher concentrations of $CO_2$ were achieved at milder temperatures. This behavior was confirmed by the study of Rodriguez Correa et al. (2018) [13], which showed that $CO_2$ is more easily produced from the organic matter at the beginning of the heating process, while hydrogen tends to be generated at higher temperatures.

The results of the total gaseous flow rate showed that the combination of both high temperature and elevated feed flow rates increased the production of gases (tests 2, 4, and 8). As expected, the highest total gaseous flow rate of 5541 N mL/min was achieved in the same operational condition where the greatest hydrogen yield was observed.

### 3.2. SCW Process Optimization: Effect of the Operational Parameters

The SCW gasification of the effluent generated during the biodiesel manufacturing process was optimized to maximize the yield of $H_2$. The volumetric flow rate, molar flow rate, and mass flow rate of hydrogen achieved at the temperature and feed flow rate conditions outlined by the composite central design are shown in Table 3.

**Table 3.** Experimental results for hydrogen production via supercritical water gasification (SCWG).

| Run | Feed Flow Rate (mL/min) | Temperature (°C) | Volumetric Flow Rate (mL/min) | Molar Flow Rate (mol/min) | Mass Flow Rate (g/min) |
|---|---|---|---|---|---|
| 1 | 12.3 | 529 | 538.9 | 22.0 | 0.044 |
| 2 | 12.3 | 671 | 2375.6 | 97.1 | 0.194 |
| 3 | 22.9 | 529 | 176.9 | 7.2 | 0.014 |
| 4 | 22.9 | 671 | 3544.5 | 144.8 | 0.290 |
| 5 | 10.0 | 600 | 871.7 | 35.6 | 0.071 |
| 6 | 25.0 | 600 | 1097.0 | 44.8 | 0.090 |
| 7 | 17.5 | 500 | 121.1 | 4.9 | 0.010 |
| 8 | 17.5 | 700 | 4092.6 | 167.2 | 0.335 |
| 9 | 17.5 | 600 | 866.6 | 35.4 | 0.071 |
| 10 | 17.5 | 600 | 1000.2 | 40.8 | 0.082 |
| 11 | 17.5 | 600 | 1003.5 | 41.0 | 0.082 |
| 12 | 17.5 | 600 | 1171.0 | 47.8 | 0.096 |

A reduced quadratic model capable of predicting the production of $H_2$ ($VH_2$; mL/min) as a function of the independent variables temperature and feed flow rate was determined according to the achieved experimental results. Equation (1) describes, in decoded values, the effect of temperature ($T$) and feed flow rate ($Qa$) on the production of $H_2$. The coefficients and the *p*-value for the analyzed variables are reported in Table S1 of Supplementary Material. The regression equation was determined considering only the significant variables of the model. In addition, reinforcing the statistical significance, the residue analysis was random and independent, with a mean of zero and constant variation, as shown in Figure S1.

$$VH_2 = 40990.58 - 581.87 \cdot Qa - 136.08 \cdot T + 0.11 \cdot T^2 + 1.01 \cdot Qa \cdot T \qquad (1)$$

ANOVA results showed a "lack-of-fit" *p*-value of 0.44, which correlates to the reliability of the model, and an $R^2$ value, indicating that 99.29% of the data variability can be expressed by the model, as shown in Table 4. This result implies the model is reasonably adequate. Thus, these results led to a greater reliability of the experimental data, guided by a statistic tool capable of systematically analyzing the behavior of the studied system.

**Table 4.** ANOVA table.

| Factor | Df | Sum Sq | Mean Sq | F Valor | *p*-Value |
|---|---|---|---|---|---|
| Feed flow rate (L) | 1 | 159,565 | 159,565 | 10.2675 | 0.049170 |
| Temperature (L) | 1 | 14,708,961 | 14,708,961 | 946.4733 | 0.000075 |
| Temperature (Q) | 1 | 2,191,096 | 2,191,096 | 140.9898 | 0.001284 |
| Two-way interaction | 1 | 582,685 | 582,685 | 37.4938 | 0.008756 |
| Lack-of-fit | 4 | 80,307 | 20,077 | 1.2919 | 0.433878 |
| Pure Error | 3 | 46,627 | 15,541 | | |
| Total SS | 11 | 17,698,445 | | | |

Note: 2 factors, 1 block, 12 runs, MS residual = 15,540.81; R-squared: 0.99283, adjusted R-squared: 0.98873.

The effects described by the regression equation (Equation (1)) were plotted into a response surface graph for easier comprehension. Figure 2 shows the effect of temperature and feed flow rate on the $H_2$ production. The results showed that temperature had a higher effect on the response when compared to the feed flow rate. At higher temperatures, an increase in feed flow rate led to a significant increase in $H_2$ production. On the other hand, at lower temperatures, an increase in the feed flow rate showed a slight tendency to decrease the analyzed response. The greater influence of temperature than the feed flow rate in relation to hydrogen production was also observed in other works reported in the literature [30,35,36].

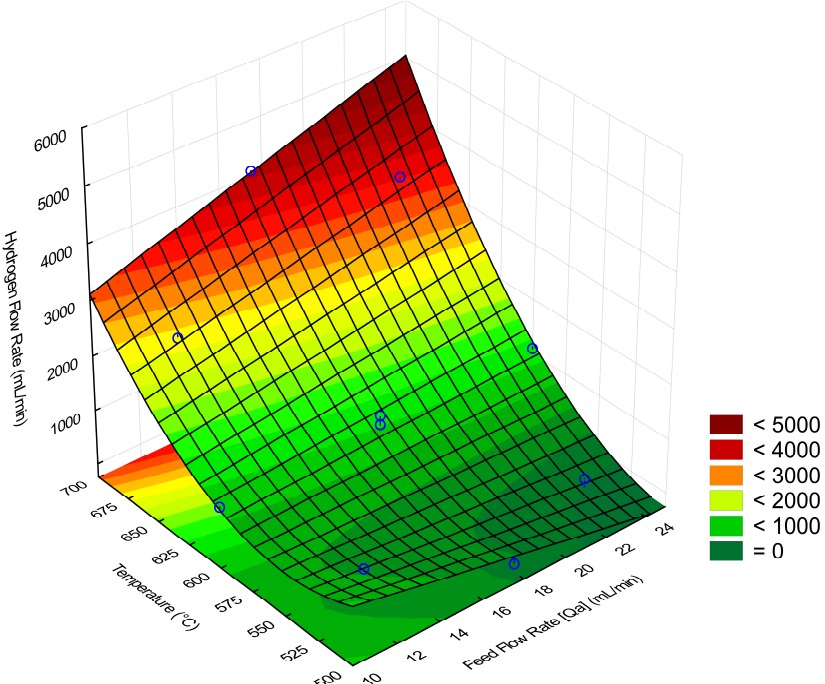

**Figure 2.** Response surface of $H_2$ production as a function of temperature and feed flow rate.

To evaluate how the hydrogen production relates to the amount of biomass processed, the ratio between the volume of $H_2$ produced and the mass of carbon fed into the SCW reactor was calculated. The results are presented in Figure 3. Among the experimental conditions assessed, the maximum $H_2$ to carbon ratio of 4.8 N mL/g of C was achieved at a temperature of 700 °C and a feed flow rate of 17.5 mL/min (run 8). The performance observed corroborates with the behavior described by the RSM graph.

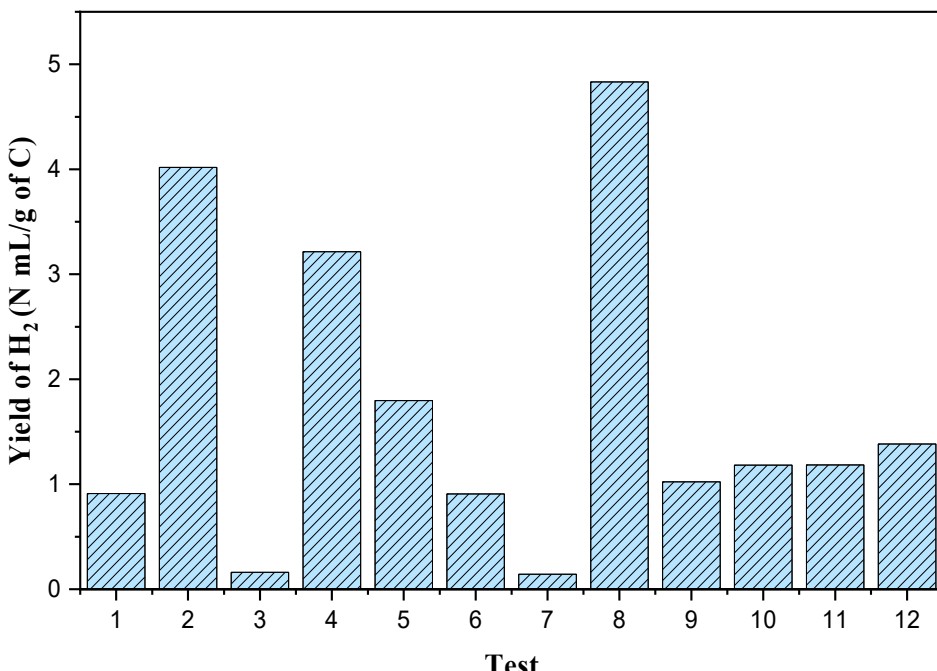

**Figure 3.** $H_2$ to carbon ratio achieved on the supercritical water gasification (SCWG) of wastewater samples from the production of biodiesel.

### 3.3. Validation of the Mathematical Regression Model

To experimentally validate the statistical regression model, an additional test was performed at a random point within the evaluated range. The random experiment was conducted at a temperature of 650 °C and a feed flow rate of 11.0 mL/min, represented in codify values as −1.23 and +0.71, respectively.

To evaluate the model, the predicted volume of $H_2$ production in random conditions was compared with the result found in the experimental test. At that condition, theoretically, the $H_2$ production should be 1554.6 mL/min, while the experimental results showed a production of 1653.6 mL/min, slightly higher than the predicted value. The error value was 5.9%, which confirms the reliability of the mathematical model. As the error was below the 10% range adopted in the CCD, the model was considered valid and able to accurately predict the behavior of the response variable for the temperature and feed flow ranges studied using this reaction unit.

### 3.4. Results of the Effluent Treatment

The main goal of this study was to assess the hydrogen production, however, the SCWG technology also offers an advantage in terms of its capacity for the simultaneous treatment of the effluent used as feedstock. In this sense, the effects of temperature and feed flow rate on the removal of total organic carbon (TOC) were also assessed and compared with the optimized conditions for hydrogen production. The obtained results are presented in detail in Table S2. In short, the response surface analysis revealed that higher TOC reduction could be achieved at a lower feed flow rate and elevated temperature, as can be seen in Figure S2. Notably, the increase in both residence time (or lower flow rates) and temperature led to greater carbon degradation/conversion, which is corroborated by the gasification results. A second-order equation of the model for TOC removal was defined as shown in Equation (2), accounting for only the statistically significant coefficients. The statistical significance of the effect estimates and the regression coefficients are reported in Tables S3 and S4, respectively.

$$R_{TOC} = 376.09 + 0.19 \cdot Qa^2 - 1.41 \cdot T + 0.001 \cdot T^2 - 0.01.01 \cdot Qa \cdot T \tag{2}$$

ANOVA results revealed an $R^2$ equal to 98.62% of the variance observed in the dataset. The "lack-of-fit" *p*-value was 0.12 and not significant, which means the adjustment and reliability of the model (Table S5). Additionally, residuals were random and independent, with a normal distribution and constant variance. The reliability of the model is evidenced by comparing the predicted and observed values, as depicted in Figure S3. TOC removal is strictly correlated with gasification extent, being both favored by higher temperatures and lower feed flow rates. Therefore, increasing the temperature and decreasing feed flow rate favors the hydrogen production as well as enhances the efficacy of effluent treatment.

TOC concentration data from untreated and treated samples were used to conduct a carbon balance calculation and assess the robustness of the experiments performed. For this purpose, the carbon content at the inlet (liquid phase) and at the outlet (comprising both liquid and gaseous phases) of the process were considered, as shown in Table 5. The mass flow rate at the inlet corresponds to the ratio between $TOC_{in}$ and feed flow rate, while the ratio between $TOC_{out}$ and flow rate is the outlet mass flow rate. Results showed a mean error below $\pm 10\%$ in an error range from 0.5 to 14.5%, which reveals the accuracy and robustness of the experimental data carried out in the SCWG system reactor.

**Table 5.** Results of carbon mass balance.

| Run | Feed Flow Rate (mL/min) | Temperature (°C) | Carbon Mass Flow Rate (g/min) | | | | |
| | | | Inlet total (Liquid Phase) | Outlet (Liquid Phase) | Outlet (Gaseous Phase) | Outlet Total | Error (%) |
| --- | --- | --- | --- | --- | --- | --- | --- |
| 1 | 12.3 | 529 | 0.592 | 0.479 | 0.129 | 0.608 | 2.7 |
| 2 | 12.3 | 671 | 0.592 | 0.103 | 0.477 | 0.580 | −2.0 |
| 3 | 22.9 | 529 | 1.104 | 1.011 | 0.099 | 1.109 | 0.5 |
| 4 | 22.9 | 671 | 1.104 | 0.471 | 0.687 | 1.158 | 4.9 |
| 5 | 10.0 | 600 | 0.485 | 0.252 | 0.209 | 0.460 | −5.1 |
| 6 | 25.0 | 600 | 1.209 | 0.997 | 0.238 | 1.215 | 0.5 |
| 7 | 17.5 | 500 | 0.848 | 0.809 | 0.162 | 0.970 | 14.5 |
| 8 | 17.5 | 700 | 0.848 | 0.220 | 0.723 | 0.942 | 11.2 |
| 9 | 17.5 | 600 | 0.848 | 0.617 | 0.161 | 0.778 | −8.2 |
| 10 | 17.5 | 600 | 0.848 | 0.611 | 0.241 | 0.852 | 0.5 |
| 11 | 17.5 | 600 | 0.848 | 0.644 | 0.171 | 0.815 | −3.8 |
| 12 | 17.5 | 600 | 0.848 | 0.630 | 0.168 | 0.798 | −5.8 |

Regarding the treatment of the wastewater generated by the biodiesel industry and used as the feedstock solution in the SCWG process for hydrogen production, a comprehensive characterization of both the untreated and treated wastewater samples was performed, as shown in Table 6. These physical-chemical parameters were used to assess the quality of the treatment process in terms of compliance with several environmental legislations. Here, it is important to highlight that these analyses were performed on samples treated under the following operational conditions: feed flow rate of 11.0 mL/min and temperature of 650 °C. These conditions were randomly selected to validate the mathematical model which was developed to determine the hydrogen production of the reaction system and were not optimized considering parameters of treatment efficiency. During the model validation experiments, which were conducted in quadruplicate, a substantial volume of treated samples was generated. These samples were subsequently utilized for the purpose of physicochemical characterization.

**Table 6.** Results of physical–chemical characterization of raw and treated wastewater samples from the biodiesel production industry.

| Parameter | Sample (mg/L) Untreated | (1) Treated | (2) Uncertainty | Legislations (3) CONAMA [37] | (4) CODEGO [38] | (5) USEPA [39] | |
|---|---|---|---|---|---|---|---|
| pH | 5.37 | 3.56 | 0.011 | 5–9 | 6–9 | 6–9 | ✘ |
| Nitrate | 1.10 | 0.20 | 0.030 | - | - | - | ✔ |
| Nitrite | 0.41 | 0.33 | 0.004 | - | - | - | ✔ |
| TOC | 48,250.00 | 20,615.00 | - | Rem. > 60% | Rem. > 60% | - | ✔ |
| COD | 339,708.00 | 89,919.00 | 0.060 | - | 1000.0 | - | ✘ |
| BOD | 141,724.00 | 30,959.50 | 0.145 | Rem. > 60% | 500.0 | 53.0 | ✘ |
| Oils and greases | 97.63 | 0.30 | 0.030 | 20.0 | 20.0 | 38.0 | ✔ |
| Sulfate | 4.4 | < | 0.010 | - | 250.0 | - | ✔ |
| Aluminum (Al) | 0.1079 | 0.1459 | 0.0023 | - | - | | ✔ |
| Barium (Ba) | 0.03666 | 0.0161 | 0.002 | 0.5 | 0.5 | 0.1040 | ✔ |
| Beryllium (Be) | 0.00555 | < | 0.001 | - | - | | ✔ |
| Calcium (Ca) | 3.5385 | 1.8652 | 0.003 | - | - | | ✔ |
| Cobalt (Co) | 0.0048 | 0.6962 | 0.004 | | | | ✔ |
| Copper (Cu) | 0.0499 | 0.1213 | 0.0007 | 1.0 | 1.0 | 0.2420 | ✔ |
| Iron (Fe) | 0.5264 | 0.3501 | 0.003 | 15.0 | 15.0 | | ✔ |
| Potassium (K) | 3.2208 | 4.3826 | 0.004 | - | - | | ✔ |
| Magnesium (Mg) | 0.8501 | 0.2616 | 0.001 | - | - | | ✔ |
| Manganese (Mn) | 0.0528 | 0.9177 | 0.003 | 1.0 | 1.0 | | ✔ |
| Molybdenum (Mo) | 0.0082 | 0.1172 | 0.003 | - | - | | ✔ |
| Sodium (Na) | 11.0503 | 10.1663 | 0.005 | - | - | | ✔ |
| Nickel (Ni) | 0.0026 | 0.2877 | 0.004 | 2.0 | 2.0 | 1.4500 | ✔ |
| Tin (Sn) | 0.5883 | 0.0064 | 0.004 | | | | ✔ |
| Zinc (Zn) | 0.0010 | 0.0290 | 0.006 | 5.0 | 5.0 | 0.4200 | ✔ |

Notes: (1) Operating conditions: 11.0 mL/min; 650 °C; and 26.0 MPa. (2) Uncertainty = expanded uncertainty (U), which is based on the combined standard uncertainty with a 95% confidence level (k = 2). (3) Conselho Nacional do Meio Ambiente—CONAMA. (4) Companhia de Desenvolvimento Econômico de Goiás—CODEGO. (5) United States Environmental Protection Agency—USEPA. Legislations compliance: green (✔) = complied with all three legislations; orange (✘) = complied with at least one legislation; red (✘) = not complied with any legislation.

The SCW process led to a TOC reduction from 48.250 to 20.6 mg/L, meaning that 57.2% of organic matter was degraded. The COD concentration decreased from 339,708 to 89,919, while the BOD concentrations reduced from 141.7 mg/L to 30.96 mg/L, representing, respectively, 73.5 and 78.1% of reduction. Most of the treatment parameters complied with the main Brazilian environmental legislation (Conselho Nacional do Meio Ambiente—CONAMA), except for pH and TOC reduction. On the other hand, the limit concentrations established by the United States Environmental Protection Agency (USEPA) and Companhia de Desenvolvimento Econômico de Goiás (CODEGO) have not been met [37–39]. Nevertheless, important progress has been made toward the treatment of the wastewater samples.

The high concentration of oils and greases (OG) in the feedstock samples was consistent with its origin: the production of biodiesel using soy oil as a feedstock material. After the treatment, a noticeable decrease in the concentration of OG (from 97.6 to 0.30 mg/L) was observed, and a value that met the legal standards for disposing of the wastewater was achieved regarding that parameter.

In supercritical conditions, a decrease in the solubility of inorganic salts and the generation of insoluble metal oxides in the supercritical medium is commonly observed. As a result, nitrate, nitrite, and sulfate concentrations decreased. The same behavior was observed for the barium, beryllium, calcium, iron, magnesium, sodium, and tin metals. In contrast, an increase in the concentration of metals such as aluminum, cobalt, molybdenum, manganese, and nickel was observed after the treatment, which could be attributed to the corrosion caused by the SCW interaction with the inner metallic alloy of the reactor wall [40]. The chemical composition of Inconel 625 is depicted in Table S6. Metals that are not inherent to the reactor's composition but exhibit an increased concentration after the reactions may be explained by the accumulation of low-solubility inorganic compounds in the supercritical medium during prolonged testing, leading to an unexpected increase in concentration during sample collection. Nevertheless, all metallic-related parameters are within the legal limits for wastewater discharge, although further work should be performed to improve this and minimize or even avoid the release of heavy metals.

### 3.5. Insights on Energy-Efficient Scale-Up

This section provides information on the scalability of the proposed SCWG system using a process simulator environment and the main energy-related aspects that should be considered. Figure 4 shows the simulation carried out in the form of a superstructure.

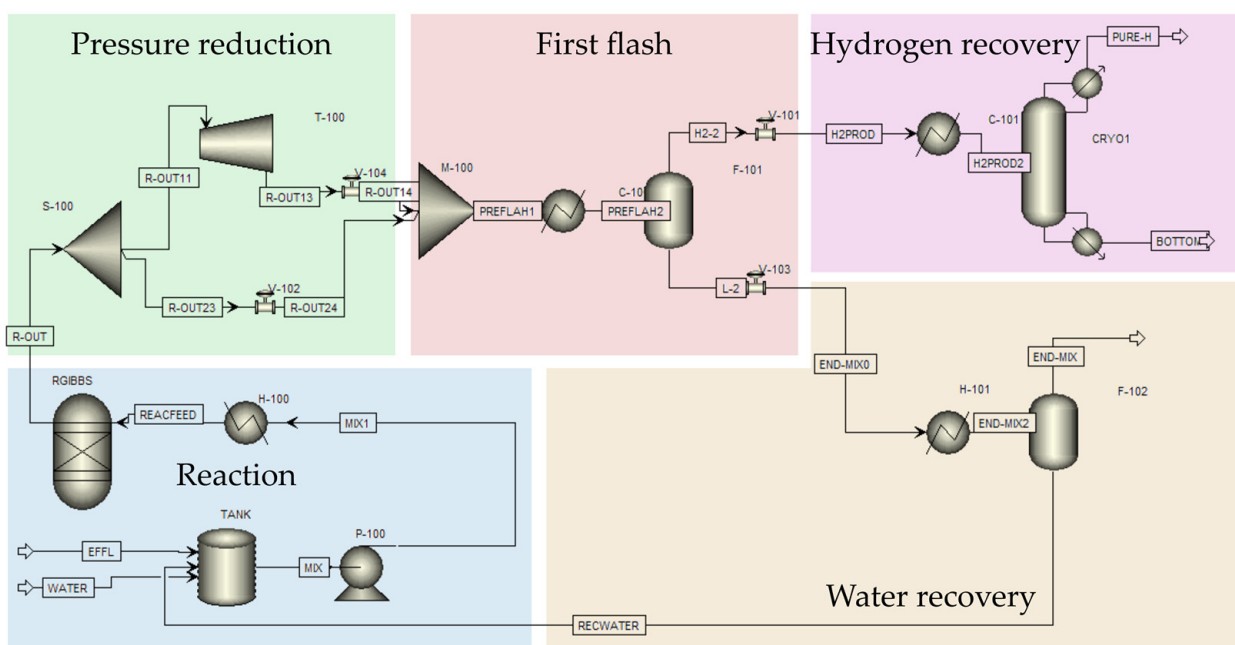

**Figure 4.** Proposed supercritical water gasification (SCWG) process of biodiesel wastewater without heat integration.

The feedstock stream (whose mass flow rate considered was 1000 kg/h) was mixed with a water stream. Therefore, the first decision variable defined in the process was the flow rate of the water stream since it directly affects the composition of the reactor's outlet stream. As in the experimental apparatus described in this paper, the feed flow rate was pressurized up to 25 MPa, then heated to 600 °C, and fed into the tubular reactor. The simulation used pump blocks and heaters to increase the pressure and temperature.

In the experimental apparatus, the products that outflow from the reactor went through cooling and depressurization through a BPR valve towards ambient conditions towards ambient conditions. Then, the products were flashed to separate the liquid and gaseous phases. In the simulation environment, it was considered that SCW processes, due to the elevated working temperature and pressure, consume a lot of energy. Therefore, the

scale-up must be carried out considering possible sources of heat and work recovery. As in the work reported by Souza et al., the options for using heat exchangers and turbines were considered [21]. The reactor outlet stream is an important source of heat and work. Options for recovering energy from this stream were considered first. Then, once the process has been defined, heat integration will be carried out.

A stream split was proposed downstream of the reactor outlet. The first fraction was directed to a turbine for power generation. The discharge pressure was set to 1.5 MPa. The stream was then depressurized to 1.0 MPa by a BPR valve, which was adequate for further separation processes and to avoid condensation in the following units. However, as this stream was expanded, its temperature was reduced, hindering its use as a heat source. Therefore, the second fraction was not expanded in a turbine. The second stream was kept available for a high-temperature heat exchange. The second flow was depressurized to 1.0 MPa by a BPR valve. Then, the split streams are mixed into a single stream cooled to 25 °C and pass through an adiabatic flash separation in unit F-101.

The liquid flow from F-101 has around 95 mol% of water. A second flash separator (F-102) operating at 50 °C was included to obtain higher purity liquid water, which could be recycled to the reaction system. The proposed configuration allows diluting 1000.0 kg/h of effluent in 10,000.0 kg/h of water with a water replacement flow of 364.5 kg/h. The steam stream from the F-102 was composed of a mixture of water, $CH_4$, and $CO_2$. The valuable $CH_4$ content in this stream (up to 179.9 kg/h from a 10:1 dilution process) could be further separated for recovery.

The F-101 gaseous stream has a high hydrogen content (up to 64.7 mol%, assuming a 10:1 dilution of water to effluent in the reactor feed). Hydrogen with market-grade purity (>99.9 mol%) can be obtained through various technologies, such as membranes, pressure swing absorption, and cryogenic separation. Most of these approaches are complex to simulate and require extensive data. For a preliminary numerical analysis, it was decided to include a cryogenic distillation system that can be readily simulated in Aspen Plus. The gaseous stream from the F-101 was cooled to a saturated liquid state to be fed into the column, which was specified for a hydrogen purity of 99.9 mol%. Initially, the column operating pressure was set to 1.0 MPa (same as the F-101 unit). However, it can be considered as a decision variable. This way, it is possible to evaluate how the pressure of the F-101 unit affects the hydrogen content in the gaseous stream and the efficiency of the subsequent separation systems.

From the defined process structure, it was necessary to identify the appropriate streams for heat recovery. There are two suitable hot streams in the process: those that split at S-100 towards the turbine and the valve (referred to as hot streams 1 and 2). Two cold flows are also present: the reactor inlet flow and the F-101 liquid flow (referred to as cold flows 1 and 2). First, a heat exchanger (E-101) can be placed between hot stream 2 (turbine outlet) and cold stream 1. This ensures the initial heating process of cold stream 1. Then, to increase the temperature of cold stream 1 even further, another heat exchanger (E-100) is placed between hot stream 1 (reactor outlet = 600 °C) and cold stream 1. Finally, to use the residual heat of hot stream 1, an additional heat exchanger can be placed to increase the temperature of cold stream 2.

It should be noted that the F-101 vapor/gaseous stream could also be considered a hot stream. However, that stream is directed to the cryogenic separation process, and its target temperature is too low for any other process stream to retrieve heat from it. Figure 5 shows the heat-integrated system.

The cryogenic process involves extremely low-temperature streams. A refrigeration cycle to provide cold utility to the cooler C-101 and the column condenser would require significant power. The turbine T-100 could be integrated into that system to at least partially provide the power required by the refrigeration cycle. This system's cost-efficiency and sustainability matters should be compared to other technologies in further works.

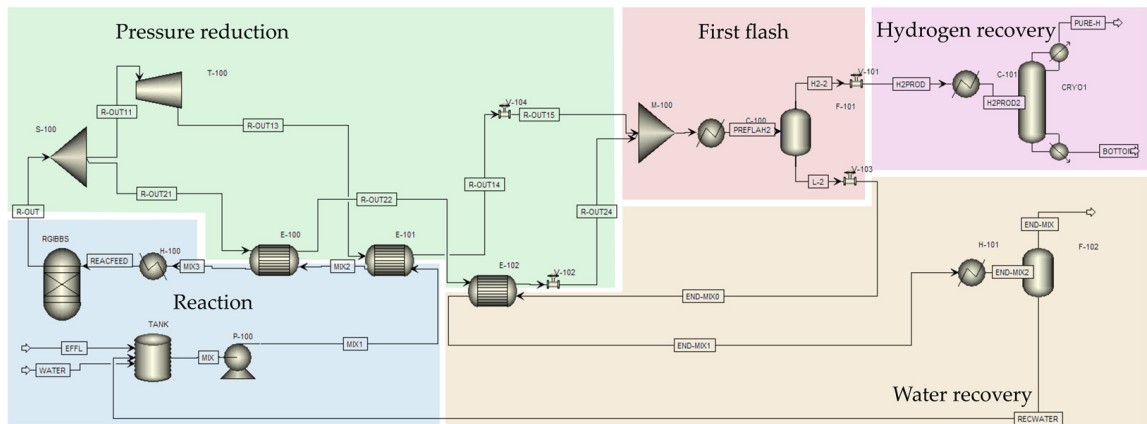

**Figure 5.** Proposed energy-integrated supercritical water gasification (SCWG) process of biodiesel wastewater.

Regarding numerical analysis, some case studies were carried out to put the simulation results into perspective. Two dilutions (water to effluent) were studied: 10:1 and 1:1. Additionally, two deviation ratios of the reactor outlet flow towards the turbine were evaluated: 0% and 50%. These combinations were investigated with and without heat integration. A total of eight case studies were proposed. Cooling for the cryogenic process was not considered in the first analysis, as it is not part of the energy integration system (its operating temperature range is too low, which forbids heat exchange with any of the other process streams). Table 7 shows the results of the simulation analysis.

**Table 7.** Case studies result in varying feed dilution, splitting ratio to turbine, and heat recovery.

| Case Study | 1 | 2 | 3 | 4 | 5 | 6 | 7 | 8 |
|---|---|---|---|---|---|---|---|---|
| Dilution | 10:1 | 10:1 | 10:1 | 10:1 | 1:1 | 1:1 | 1:1 | 1:1 |
| Splitting ratio to turbine | 0.0 | 0.0 | 0.5 | 0.5 | 0.0 | 0.0 | 0.5 | 0.5 |
| E-100 duty (kW) | 0.0 | 8100.0 | 0.0 | 4100.0 | 0.0 | 1100.0 | 0.0 | 500.0 |
| E-101 duty (kW) | 0.0 | 0.0 | 0.0 | 1500.0 | 0.0 | 0.0 | 0.0 | 300.0 |
| E-102 duty (kW) | 0.0 | 364.7 | 0.0 | 364.7 | 0.0 | 49.5 | 0.0 | 49.5 |
| $H_2$ in reactor outlet (kmol/h) | 31.32 | 31.32 | 31.32 | 31.32 | 4.856 | 4.856 | 4.856 | 4.856 |
| $H_2$ mole fraction in reactor outlet | 0.0507 | 0.0507 | 0.0507 | 0.0507 | 0.0490 | 0.0490 | 0.0490 | 0.0490 |
| $H_2$ in F-101 vapor (kmol/h) | 30.98 | 30.98 | 30.98 | 30.98 | 4.849 | 4.849 | 4.849 | 4.849 |
| $H_2$ mole fraction in F-101 vapor | 0.6475 | 0.6475 | 0.6475 | 0.6475 | 0.1232 | 0.1232 | 0.1232 | 0.1232 |
| H-100 duty (kW) | 10,733.2 | 2633.2 | 10,733.2 | 5133.2 | 1623.3 | 523.3 | 1623.3 | 823.3 |
| H-101 duty (kW) | 364.7 | 0.0 | 364.7 | 0.0 | 49.5 | 0.0 | 48.5 | 0.0 |
| C-100 duty (kW) | −9749.2 | −1649.24 | −8834.3 | −3234.3 | −1241.6 | −141.6 | −1084.3 | −234.0 |
| T-100 work (kW) | 0.0 | 0.0 | −914.9 | −914.9 | 0.0 | 0.0 | −157.3 | −157.3 |

A few points are worth highlighting: (i) concerning the dilution adopted and how it affects the process energetically, energy consumption does not increase in the same proportion as dilution. For example, in the case of 10:1 dilution without a turbine (case study 1), the heating energy required in the reactor preheater is 6.6 times greater than in the 1:1 system. In the case of heat integration, this ratio becomes 5.0; (ii) the hydrogen content of the reactor outlet flow is considerably higher when the reaction is conducted at a higher dilution (confirming that the water itself acts both as reaction media and catalyst in the SCWG process). After F-101, a flow of 64.7 mol% is obtained in the 10:1 case compared to 12.3 mol% in the 1:1 case; (iii) heat integration increases the energy efficiency of the system.

However, systems with turbine power generation have considerably less energy available for heat exchange. These trade-offs must be evaluated carefully since electricity and utility costs can be highly uncertain.

Additionally, case study 4 (10:1 dilution, heat recovery, and 50% turbine bypass) was considered to analyze the different flash pressures and how the purification system was affected. In the cryogenic column, the RadFrac model was used with a partial condenser, 1.0 MPa pressure (no pressure drops in the column), 10 stages, feed in stage 5, reflux ratio of 1.0, and hydrogen purity of 99.9 mol% in the distillate stream. The flash pressure (F-101) was then varied to assess how other separation systems were affected. Table 8 shows the main results obtained show a varying flash pressure.

**Table 8.** Case studies varying flash pressure.

| Case Study | 4.1 | 4.2 | 4.3 |
|---|---|---|---|
| F-101 pressure (MPa) | 1.0 | 3.0 | 5.0 |
| Distillate rate (kmol/h) | 31.01 | 30.20 | 29.31 |
| Feed $H_2$ mole flow (kmol/h) | 30.98 | 30.17 | 29.29 |
| Feed $H_2$ composition | 0.6475 | 0.8273 | 0.8805 |
| Dist. $H_2$ mole flow (kmol/h) | 30.98 | 30.17 | 29.29 |
| Dist. $H_2$ composition | 0.999 | 0.999 | 0.999 |
| T-100 work (kW) | −914.9 | −738.2 | −590.2 |
| C-101 duty (kW) | −188.3 | −110.6 | −91.8 |
| Condenser duty (kW) | −110.1 | −93.6 | −96.2 |
| Reboiler duty (kW) | 136.0 | 105.3 | 96.0 |

Notably, increasing the pressure in the flash tank results in lower energy consumption in the additional separation unit. The total energy related to cooling (C-101 and condenser duties) decreases considerably at high pressures. The F-101 steam becomes purer in hydrogen at high pressures. However, some compensations must be considered: hydrogen recovery decreases because, as the pressure increases, more hydrogen leaves the flash unit diluted in the liquid stream; work generation in the T-100 also decreases due to the higher discharge pressures.

The proposed methodology does not provide a definitive project due to uncertainties in operating and capital costs. At this point, it should be recalled that the assessment of energy required for scale-up was based on the effluent characterized by the work of Yu et al. (2021) [34]. Therefore, overall, the composition of the effluent can vary, requiring operational flexibility in the process. These uncertainties require further research that are beyond the scope of this study. The proposed methodology employs a superstructure model, adaptable to various needs, highlighting energy-efficient unit operations and potential operating conditions. The computational experiments were required to evaluate energy consumption and product compositions after selecting the decision variables.

## 4. Conclusions

The supercritical water gasification of the wastewater generated in a biodiesel industry showed promising results in producing hydrogen. At optimized conditions, that is, a temperature of 600 °C and a feed flow rate of 17.5 mL/min (run 8), a total gas flow rate of 5541 mL/min was achieved. Hydrogen represented 73.86% of the gaseous products. It was observed that both temperature and feed flow rate positively influence the production of hydrogen via the SCWG. The effect was more pronounced for the temperature parameter when compared to the feed flow rate. A reduced quadratic model capable of predicting the production of $H_2$ as a function of the independent variables, temperature, and feed flow rate, was determined with the aid of the Statistica™ software version 12. The statistical

model was validated by comparing the model-predicted values with those obtained experimentally. A coefficient of determination ($R_2$) above 0.99 and an error smaller than 10% were observed, proving that the model could perform a predictive analysis of the behavior of $H_2$ production for the range of temperature and feed flow rate studied. Additionally, an extensive physicochemical characterization of the effluent samples before and after the SCW process was performed. The results were compared to different environmental legislation, and significant progress on the treatment of the effluent was observed, although further improvements are to be implemented for the sake of an increasing enhancement with time of the feasibility and sustainability of the studied technology towards its final commercial development. Finally, information on the scalability of the proposed SCWG system considering the main energy-related aspects was provided.

**Supplementary Materials:** The following supporting information can be downloaded at: https://www.mdpi.com/article/10.3390/w15234062/s1, Figure S1: (a) Distribution of residual; (b) Residuals x predicted values; Figure S2: Response surface of TOC removal as a function of temperature and feed flow rate; Figure S3: (a) Distribution of residual; (b) Residuals x predicted values; Table S1: Effect estimates in SCWG for hydrogen production; Table S2: Removal of total organic carbon after SCWG process; Table S3: Effects estimate in SCWG for removal of total organic carbon; Table S4: Regression coefficients in SCWG for removal of total organic carbon; Table S5: ANOVA table in SCWG for removal of total organic carbon; Table S6: Chemical composition of Inconel 625.

**Author Contributions:** I.R.T.: methodology, formal analysis, data curation, and writing—original draft. I.M.D.: validation, investigation, formal analysis, and writing. L.C.M.: validation, investigation, and formal analysis. L.A.A.: investigation and writing. L.V.P.: software and writing—original draft. G.B.M.S.: methodology, formal analysis, and writing—review and editing. L.C.-F.: supervision, funding acquisition—review and editing. C.G.A.: supervision, project administration, funding acquisition—review and editing. J.M.A.-P.: formal analysis and writing. R.G.: supervision, funding acquisition—review. All authors have read and agreed to the published version of the manuscript.

**Funding:** The authors gratefully acknowledge the financial support from Conselho Nacional de Desenvolvimento Científico e Tecnológico—CNPq (grants #407158/2013-8, #431642/2016-8, and #405851/2022-7). This study was financed in part by the Coordenação de Aperfeiçoamento de Pessoal de Nível Superior—CAPES—Brasil—Finance Code 001.

**Data Availability Statement:** All data are contained within the article and Supplementary Material.

**Acknowledgments:** The authors would like to thank the following partners, the Lab Aqualit Tecnologia em Saneamento Ltda.

**Conflicts of Interest:** The authors declare that they have no known competing financial interest or personal relationships that could have appeared to influence the work reported in this paper.

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
