# Peer review of "Continuous Hydrogen Production via Hydrothermal Gasification of Biodiesel Industry Wastewater: Experimental Optimization and Energy Integration Simulation"

_water, doi:10.3390/w15234062_

Round 1

Reviewer 1 Report

Comments and Suggestions for Authors

The proposed manuscript concerned a new hydrogen production approach from the wastewater using supercritical flow reactor. The authors thoroughly studied a wide range of possible processing conditions experimentally as well as using computer simulation. A perspective of production scaling was estimated. At the moment, this approach has its own disadvantages, but they were mentioned and discussed by the authors. In my opinion, the manuscript could be considered for publication in Water after some improvement. Please find the following recommendations.

1)     The SCWG system in the presented work was aimed to transform the wastewater, which contents could differ with the time, season, etc. It would be helpful for further development of the proposed technique to estimate the reproducibility of hydrogen production by SCWG in some fixed conditions;

2)     In the Introduction and/or Discussion sections, the authors should mention different methods currently known for hydrogen production from wastewater, for example, fermentation, photocatalysis, electrochemical processes and so on. Please show the competitiveness of the proposed SCWG process to those techniques;

3)     From the experiment description, it remained not quite clear why the contents of potassium increased in water after gasification? Also, could the authors provide the composition of the reactor material to compare it with the list of ions, which concentrations increased after the processing.

4)     The manuscript should be checked for typos. For example, in lines 166, 356-357.

Comments on the Quality of English Language

Minor corrections mostly concerning the typos are required.

Author Response

Reviewer #1:

General comment:

The proposed manuscript concerned a new hydrogen production approach from the wastewater using supercritical flow reactor. The authors thoroughly studied a wide range of possible processing conditions experimentally as well as using computer simulation. A perspective of production scaling was estimated. At the moment, this approach has its own disadvantages, but they were mentioned and discussed by the authors. In my opinion, the manuscript could be considered for publication in Water after some improvement. Please find the following recommendations.

Answer: Thank you. We appreciate the good evaluation of our work.

Specific comments:

1) The SCWG system in the presented work was aimed to transform the wastewater, which contents could differ with the time, season, etc. It would be helpful for further development of the proposed technique to estimate the reproducibility of hydrogen production by SCWG in some fixed conditions;

Answer: Thank you. We agree that reproducibility is an important aspect of scientific research, and we acknowledge the importance of establishing the reliability of our proposed technique. Nevertheless, our process has proven to be adaptable to different types of raw materials, proving effective for the gasification of organic matter and satisfactory hydrogen production, as can be seen in works from our own group applying supercritical technology in the gasification of hormones (DOI 10.1016/j.jece.2021.106095) and antibiotics (DOI  10.1016/j.watres.2023.119826) present in pharmaceutical industry wastewater, and gasification of glycerol originating from the biodiesel industry wastewater (DOI 1016/j.ijhydene.2023.04.008), which have already been mentioned in this work.

2) In the Introduction and/or Discussion sections, the authors should mention different methods currently known for hydrogen production from wastewater, for example, fermentation, photocatalysis, electrochemical processes and so on. Please show the competitiveness of the proposed SCWG process to those techniques;

Answer: Thank you. We appreciate the importance of contextualizing our work within the broader landscape of hydrogen production from wastewater. An overview of the different methodologies was added as follows:

Page 2 – Lines 57 – 74

“According to Rodriguez Correa & Kruse (2018) [13], several methods are widely used to obtain hydrogen. The more traditional ones are based on techniques such as steam reforming, pyrolysis, and partial oxidation. Additionally, biological, and electrochemical routes are also studied for hydrogen production as alternative methods[14]. […] Industrial by-products or even wastes can be understood as attractive sources of organic matter for hydrogen production [17–20]. Zoppi and co-workers (2022) reported a carefully review study using aqueous phase reforming process for the valorization of wastewater streams. Nonetheless, low carbon conversion and total organic carbon degradation were observed. Such issues could be overcome by using both higher pressure and temperature conditions [21]. Therefore, the possibility of using different raw materials coupled with the high reaction velocity and conversion, highlights the SCWG potential in comparison to other alternative techniques.”

3) From the experiment description, it remained not quite clear why the contents of potassium increased in water after gasification? Also, could the authors provide the composition of the reactor material to compare it with the list of ions, which concentrations increased after the processing.

Answer: We appreciate your attention to detail. Regarding the increase in potassium content in water after gasification, a plausible explanation arises from specific conditions in the reaction system. As inorganic species are insoluble in water under supercritical conditions, there is a possibility of the accumulation of these species inside the reactor during extended reaction tests. For a complete characterization of the effluent post-reactor, an extended reaction time was employed to accumulate the minimum required volume for all analytical methodologies. Consequently, a continuous accumulation of certain species in the system over the course of the tests could result in an undue concentration at a specific sample collection, which leads to an experimental error.

The hypothesis of potassium ion increasing from the reactor's metal alloy is unlikely, given that the Inconel alloy typically consists of the metallic elements Ni, Cr, Mo, and Fe, (in descending order of content) and minor fractions of Mn, Al, and Co (as presented in Table S6 in supplementary materials). Thus, the increase in some metallic species such as Ni, Mo, and Mn, notwithstanding small, is explained by some leaching of material from the reaction system.

We made improvements in the manuscript addressing this point raised by the referee.

Page 11, Lines 352 – 363

At supercritical conditions is commonly observed a decrease in the solubility of inorganic salts and the generation of insoluble metal oxides in the supercritical medium. As a result, nitrate, nitrite, and sulfate concentrations decreased. The same behavior was observed for the barium, beryllium, calcium, iron, magnesium, sodium, and tin metals. In contrast, an increase in the concentration of metals such as aluminum, cobalt, molybdenum, manganese, and nickel were observed after the treatment, which could be attributed to the corrosion caused by the SCW interaction with the inner metallic alloy of the reactor wall [40]. The chemical composition of Inconel 625 is depicted in Table S6. Metals that are not inherent to the reactor's composition but exhibit an increased concentration after the reactions may be explained by the accumulation of low-solubility inorganic compounds in the supercritical medium during prolonged testing, leading to an unexpected increase in concentration during sample collection. Nevertheless, all metallic related parameters are within the legal limits for wastewater discharge, although further work should be performed to improve this and minimize or even avoid the release of heavy metals.

4) The manuscript should be checked for typos. For example, in lines 166, 356-357.

Answer: Thank you. The entire document was revised.

Page 5 – Lines 173 – 175

[…] For this purpose, the species considered during the simulation of the SCW biodiesel wastewater processing were based on the effluent characterized by the work […]

Reviewer 2 Report

Comments and Suggestions for Authors

In the present work, Teixeira et al. studied the hydrothermal gasification of biodiesel-derived wastewater, with the aim of producing hydrogen. The topic is of interest to the readers of the journal. The manuscript is well written, the novelty is clearly explained and scientifically sound. In my opinion, it can be accepted for publication after that the following points will be properly addressed

·         Row 48: please use more updated data for the energy mix; six years is a long timeframe, especially in the current fast changing scenario

·         In the introduction, a wider view on the possible valorization pathway for biodiesel-derived wastewater should be provided. For example, aqueous phase reforming may be mentioned, being an alternative option to the technology proposed herein (see 10.1016/j.cattod.2021.06.002 and 10.1016/j.jclepro.2023.138141)

·         Row 166: the authors used a likely different composition (the one from ref 28) to model the performance of their own technology; despite I understand the effort, I suggest to add the limitations for this approach.

·         In paragraph 2, the information regarding the carbon balance should be added to check the robustness of the experiments

·         I suggest modifying the legend in Figure 2 (e.g., formally everything is < 5000 and hence should be the same colour)

·         Paragraph 3.4: interestingly, there is a trade off between the conditions which maximize the hydrogen production goal and the ones for TOC removal goal, which is important as well. The authors should elaborate more on this point, since it can be discussed if one of them should be chosen

·         I suggest improving the quality of Figure 5, and maybe have a new screenshot, since in the current file the gas-liquid separator is highlighted

Author Response

Reviewer #2:

General comment:

In the present work, Teixeira et al. studied the hydrothermal gasification of biodiesel-derived wastewater, with the aim of producing hydrogen. The topic is of interest to the readers of the journal. The manuscript is well written, the novelty is clearly explained and scientifically sound. In my opinion, it can be accepted for publication after that the following points will be properly addressed.

Answer: Thank you for your positive judgment.

Specific comments:

1)·Row 48: please use more updated data for the energy mix; six years is a long timeframe, especially in the current fast changing scenario.

Answer: Data for the energy mix was actualized as can be seen below:

Page 2 – Lines 48 – 50

“Considering the current sources of energy supply, in 2022, the fossil fuels represented about 60.93% of the total generation, while renewable and nuclear sources accounted, respectively, for 29.92% and 9.15% [7].”

2) In the introduction, a wider view on the possible valorization pathway for biodiesel-derived wastewater should be provided. For example, aqueous phase reforming may be mentioned, being an alternative option to the technology proposed herein (see 10.1016/j.cattod.2021.06.002 and 10.1016/j.jclepro.2023.138141)

Answer: Thanks. References suggested were added in the manuscript.

Page 2 – Lines 58 – 60

“According to Rodriguez Correa & Kruse (2018), several methods are widely used to obtain hydrogen [13]. The more traditional ones are based on techniques such as steam reforming, pyrolysis, and partial oxidation [14–16]. Additionally, biological, and electrochemical routes were also studied for hydrogen production as alternative methods [17].”

Page 2 – Lines 69 – 75

“Industrial by-products or even wastes can be understood as attractive sources of organic matter for hydrogen production [17–20] Zoppi and co-workers (2022) reported a carefully review study using aqueous phase reforming process for the valorization of wastewater streams. Nonetheless, low carbon conversion and total organic carbon degradation were observed. Such issues could be overcome by using both higher pressure and temperature conditions [21]. Therefore, the possibility of using different raw materials coupled with the high reaction velocity and conversion, highlights the SCWG potential in comparison to other alternative techniques.”

3) Row 166: the authors used a likely different composition (the one from ref 28) to model the performance of their own technology; despite I understand the effort, I suggest to add the limitations for this approach.

Answer: Thanks. We made improvements in the manuscript addressing this point raised by the referee.

Page 14 – Lines 481 – 486

“The proposed methodology does not provide a definitive project due to uncertainties in operating and capital costs. At this point, it should be recalled that the assessment of energy required for scale-up was based on the effluent characterized by the work of Yu et al. (2021) [40]. Therefore, overall, the composition of the effluent can vary, requiring operational flexibility in the process. These uncertainties require further research that are beyond the scope of this study.”

4) In paragraph 2, the information regarding the carbon balance should be added to check the robustness of the experiments

Answer: Data on carbon mass balance were provided.

Page 9 – Lines 298 – 305;

“TOC concentration data from untreated and treated samples were used to conduct a carbon balance calculation and assess the robustness of the experiments performed. For this purpose, the carbon content at the inlet (liquid phase) and at the outlet (comprising both liquid and gaseous phases) of the process were considered, as shown in Table 5. The mass flow rate at the inlet corresponds to the ratio between TOCin and feed flow rate, while the ratio between TOCout and flow rate is the outlet mass flow rate. Results showed a mean error below ± 10% in an error range from 0.5 to 14.5%, which reveals the accuracy and robustness of the experimental data carried out in the SCWG system reactor.”

Page 9 – Table 5

Table 5. Results of carbon mass balance

Run

Feed flow rate (mL/min)

Temperature (°C)

Carbon mass flow rate (g/min)

Inlet total

(liquid phase)

Outlet

(liquid phase)

Outlet

(gaseous phase)

Outlet

total

Error

(%)

1

12.3

529

0.592

0.479

0.129

0.608

2.7

2

12.3

671

0.592

0.103

0.477

0.580

-2.0

3

22.9

529

1.104

1.011

0.099

1.109

0.5

4

22.9

671

1.104

0.471

0.687

1.158

4.9

5

10.0

600

0.485

0.252

0.209

0.460

-5.1

6

25.0

600

1.209

0.997

0.238

1.215

0.5

7

17.5

500

0.848

0.809

0.162

0.970

14.5

8

17.5

700

0.848

0.220

0.723

0.942

11.2

9

17.5

600

0.848

0.617

0.161

0.778

-8.2

10

17.5

600

0.848

0.611

0.241

0.852

0.5

11

17.5

600

0.848

0.644

0.171

0.815

-3.8

12

17.5

600

0.848

0.630

0.168

0.798

-5.8

5) I suggest modifying the legend in Figure 2 (e.g., formally everything is < 5000 and hence should be the same colour)

Answer: Thanks. We understand you concern, but it is commonly reported this way. Furthermore, we believe that this will not cause misunderstandings for readers.

6) Paragraph 3.4: interestingly, there is a trade off between the conditions which maximize the hydrogen production goal and the ones for TOC removal goal, which is important as well. The authors should elaborate more on this point, since it can be discussed if one of them should be chosen.

Answer: The main goal of this study was the SCWG process optimization to maximize hydrogen production. In this sense, optimized conditions reported were based on this purpose. Nonetheless, optimized conditions for total organic carbon removal were also evaluated and compared with optimized conditions for hydrogen production. Results showed that optimized condition was favorable for both total organic carbon and gasification. To make it clear improvements were performed in the manuscript as highlighted bellow:

Page 8 and 9 – Lines 276 – 297

“The main goal of this study was to assess the hydrogen production, however, the SCWG technology also offers an advantage in terms of its capacity for the simultaneous treatment of the effluent used as feedstock. In this sense, the effects of temperature and feed flow rate on the removal of total organic carbon (TOC) were also assessed and compared with the optimized conditions for hydrogen production.

[…]

TOC removal is strictly correlated with gasification extent, being both favored by higher temperatures and lower feed flow rates. Therefore, increasing the temperature and decreasing feed flow rate favors the hydrogen production as well as enhances the efficacy of effluent treatment.”

7) I suggest improving the quality of Figure 5, and maybe have a new screenshot, since in the current file the gas-liquid separator is highlighted.

Answer: Figure quality was improved.
